# Understanding patient preferences in anti-VEGF treatment options for age-related macular degeneration

**Semra Ozdemir**[1,2]\*, **Eric Finkelstein**[1,2,3], **Jia Jia Lee**[1], **Issac Horng Khit Too**[4], **Kelvin Yi Chong Teo**[5,6], **Anna Chen Sim Tan**[5,6], **Tien Yin Wong**[5,6], **Gemmy Chui Ming Cheung**[5,6]

**1** Signature Programme in Health Services and Systems Research, Duke-NUS Medical School, Singapore, Singapore, **2** Saw Swee Hock School of Public Health, National University of Singapore, Singapore, Singapore, **3** Duke Global Health Research Institute, Duke University, Durham, North Carolina, United States of America, **4** Medical Affairs, Novartis Singapore Private Limited, Singapore, Singapore, **5** Medical Retina, Ophthalmology, Singapore National Eye Centre, Singapore, Singapore, **6** Ophthalmology & Visual Sciences Academic Clinical Program, SingHealth Duke-NUS, Singapore, Singapore

\* semra.ozdemir@duke-nus.edu.sg

## Abstract

### Purpose

(1) To investigate the relative importance of convenience (consultation frequency and injection frequency) against treatment outcomes (visual and anatomical outcomes) and out-of-pocket medical costs via a discrete choice experiment (DCE), and (2) to investigate how patient characteristics affect patient treatment preferences.

### Methods

Eligibility criteria were: (1) receiving a neovascular age-related macular degeneration (nAMD) diagnosis; (2) receiving anti-VEGF treatment; (3) being ≥21 years old, and (4) being able to speak and understand English/Mandarin. Patients were presented with eight choice tasks and asked to choose between their current treatment and two hypothetical treatments that varied by six attributes: number of clinic visits in a year, number of injections in a year, vision quality, control of swelling in retina, drug labelling and out-of-pocket cost.

### Results

This analysis involved 180 patients. Based on latent class logistic regressions, vision quality was the most important attribute (34%) followed by cost (24%). The frequency of total clinic visits (15%) was the third most-important attribute, closely followed by labelling (12%) and control of retina swelling (11%). Injection frequency was the least important attribute (4%).

### Conclusions

Vision quality was the most important attribute followed by the out-of-pocket costs. Given the same outcomes, patients preferred treatment regimens which require fewer total clinic visits. In comparison, injection frequency alone did not influence patient preferences. With

---

**Data Availability Statement:** As approved by the ethics committees (SingHealth Centralised Institutional Review Board (CIRB Ref.: 2019/2346) and National University of Singapore-Institutional

Review Board (NUS-IRB Ref: N-19-063), the data is only accessible to the study team members. However, de-identified data may be shared upon reasonable request. Every request will be reviewed by the approving Institutional Review Board (SingHealth Centralised Institutional Review Board: irb@singhealth.com.sg, +65 6323 7515; National University of Singapore-Institutional Review Board: irb@nus.edu.sg, +65 6516 4311) and the researcher will need to sign a data access agreement with National University of Singapore after approval.

**Funding:** This study was funded by Novartis (Singapore), grant number: R-913-301-510-592. The funder participated in the design of the study, the approval of the manuscript, and the decision to submit the manuscript for publication. The funder had no role in the collection, management, analysis, and interpretation of the data; preparation, and review of the manuscript.

**Competing interests:** Semra Ozdemir, Anna Cheng Sim Tan, Kelvin Yi Chong Teo, Tien Yin Wong and Gemmy Chui Ming Cheung reported receiving grants from Novartis during the conduct of the study. Issac Horng Khit Too is being employed by Novartis. Eric Finkelstein and Jia Jia Lee reported no conflict of interest. This does not alter our adherence to PLOS ONE policies on sharing data and materials.

increasing treatment options for nAMD, understanding patients' preferences can help clinicians in selecting agents and treatment regimen most preferred for each patient, which may lead to improved long-term adherence and outcomes.

## Introduction

Age-related macular degeneration (AMD) is one of the leading causes of irreversible vision loss among the elderly worldwide, accounting for 5% of global blindness [1]. With an aging global population, AMD prevalence is projected to increase by 20% from 195.6 million in 2020 to 243.3 million by 2030 [2,3]. Neovascular AMD (nAMD), an advanced stage of AMD, is responsible for the majority of AMD-related blindness. For the management of nAMD, the classic monthly or bi-monthly (fixed) dosing of the intravitreal injections of anti–vascular endothelial growth factor (anti-VEGF) has been proven efficacious and safe in clinical trials [4,5] and has been used as standard care for nAMD in most countries [6,7]. However, monthly or bi-monthly injections can be unsustainable in many real-world clinical practices. A more flexible approach, *pro-re-nata* (PRN) dosing (treat based on certain visual and anatomical criteria) has demonstrated comparable outcomes at 12 months but the visual gain is not sustained in the long term [8,9]. The treat-and-extend approach (treating at every clinic visit where visits are extended until an ideal interval is established for each patient) has demonstrated good visual outcomes with fewer injections and/or clinical visits [10,11] and has the potential to be more convenient and less expensive to patients and healthcare systems.

Multiple anti-VEGF agents are increasingly becoming available for nAMD treatment, with newer agents (e.g., broluciuzmab, faricimab) potentially offering longer durability alongside fewer injections [12–15]. These treatment regimens impose different logistical and/or financial burden to patients and their informal caregivers who often accompany patients to the clinic visits [16–18]. Therefore, for effective AMD management, it is important to understand patient treatment preferences and to what extent patients trade-off convenience factors, such as frequency of clinic visits and injections, with therapeutic benefits and medical costs. A state-of-the-art method that has been extensively used to measure preferences for healthcare products and services is the Discrete Choice Experiment (DCE). It is a survey research method which assesses how patients trade-off between different attributes of healthcare products and services by asking patients to choose their preferred choice between two or more alternatives [19].

Few studies have used DCEs to investigate how nAMD patients weigh treatment outcomes against convenience factors (e.g., consultation and injection frequency). Findings from these studies were also contradictory. While results from one study [20] indicated that patients preferred fewer consultations and injections (i.e. treat-and-extend), another [21] demonstrated that patients preferred frequent monitoring but fewer injections (i.e. PRN regimen). Yet another [22] found that the frequency of consultations and injections did not significantly affect preferences. Two of these studies [20,22] also found that vision-quality improvements were more important than convenience factors. Out-of-pocket costs, a variable highly correlated with injection and consultation frequency in healthcare systems where users pay out-of-pocket for these services [23–25], have not been examined in any of the aforementioned studies [20–22]. A recent paper [26] investigated the importance of cost to both insurance providers and nAMD patients and found that vision quality and cost to the patient were the two most important attributes while cost to the insurance provider was the least important. Our study investigated the relative importance of convenience (consultation frequency and injection

frequency) against treatment outcomes (visual and anatomical outcomes) and out-of-pocket medical costs via a DCE. The secondary aim was to investigate how patient characteristics such as age, years on treatment, attitudes toward injections, affordability, and visual-related quality of life affect patient treatment preferences. The results of this study may inform clinicians on patient preferences to facilitate shared decision-making, and healthcare policymakers for approval of new drugs.

## Methods

### Study setting and participants

This cross-sectional study took place at the Singapore National Eye Centre between September 2019 and January 2021. Data collection was halted temporarily between February 2020 and June 2020 due to COVID-19 restrictions. Eligible patients: (1) had a nAMD diagnosis; (2) were currently undergoing anti-VEGF treatment; (3) were at least 21 years old and (4) were able to speak and understand English or Mandarin. The study was approved by the SingHealth Centralised Institutional Review Board (CIRB Ref.: 2019/2346) and National University of Singapore-Institutional Review Board (NUS-IRB Ref: N-19-063). Eligible patients were identified from medical records. Trained interviewers approached eligible patients and obtained written informed consent. The survey was administered face-to-face via the Qualtrics platform using a tablet. Patients answered the survey in English or Mandarin depending on individual preference. All patients were reimbursed upon study completion. Of the 236 eligible patients identified, 52 declined to participate. The remaining 184 who agreed provided written informed consent and completed the survey (119 in English and 65 in Mandarin) (Fig 1). Four patients were excluded from the final analytic sample as they had participated in pre-test interviews.

### Establishing attributes and levels

DCEs use a series of choice tasks where individuals select the preferred alternative from two or more alternatives with selected attributes [19]. These attributes are characterized by their levels. The utility that the individuals achieve is determined by the different attribute level combinations.

Attributes and levels that could affect patient preferences for nAMD treatment were identified via literature reviews and consultation with the clinical experts. The initial draft was pre-tested with nine participants using the think-aloud technique. The pre-test interviews investigated the appropriateness, feasibility, and relevance of the included attributes. The survey was revised based on the participant feedback.

The final attributes included were (1) Number of total clinic visits in a year (6/9/12 visits in a year); (2) Number of injections in a year (4/6/9/12 injections in a year); (3) Vision quality (good, moderate or poor); (4) Control on the swelling in retina (well-controlled swelling, moderately-controlled swelling or poorly-controlled swelling); (5) Drug labelling (on-label or off-label); and (6) Annual out-of-pocket cost for managing nAMD. Table 1 presents the definition of attributes and levels. The levels for the cost attribute were selected based on the medication costs (both unsubsidized and subsidized by the government based on means-testing) at local hospitals (which can range between SGD150-1,500 (~USD108-1080) per injection). To reduce cognitive burden, patients were shown the total annual cost (based on the number of injections times the cost of injection). Treatment profiles were created such that the injection frequency was not more than the total number of clinic visits. In addition, a treatment profile with 'good vision quality' did not have 'poorly-controlled swelling in the retina' since this was not realistic. Illustrations were used to explain the attributes where necessary. In each DCE task, participants were first asked to choose between two hypothetical treatments (Treatment A vs.

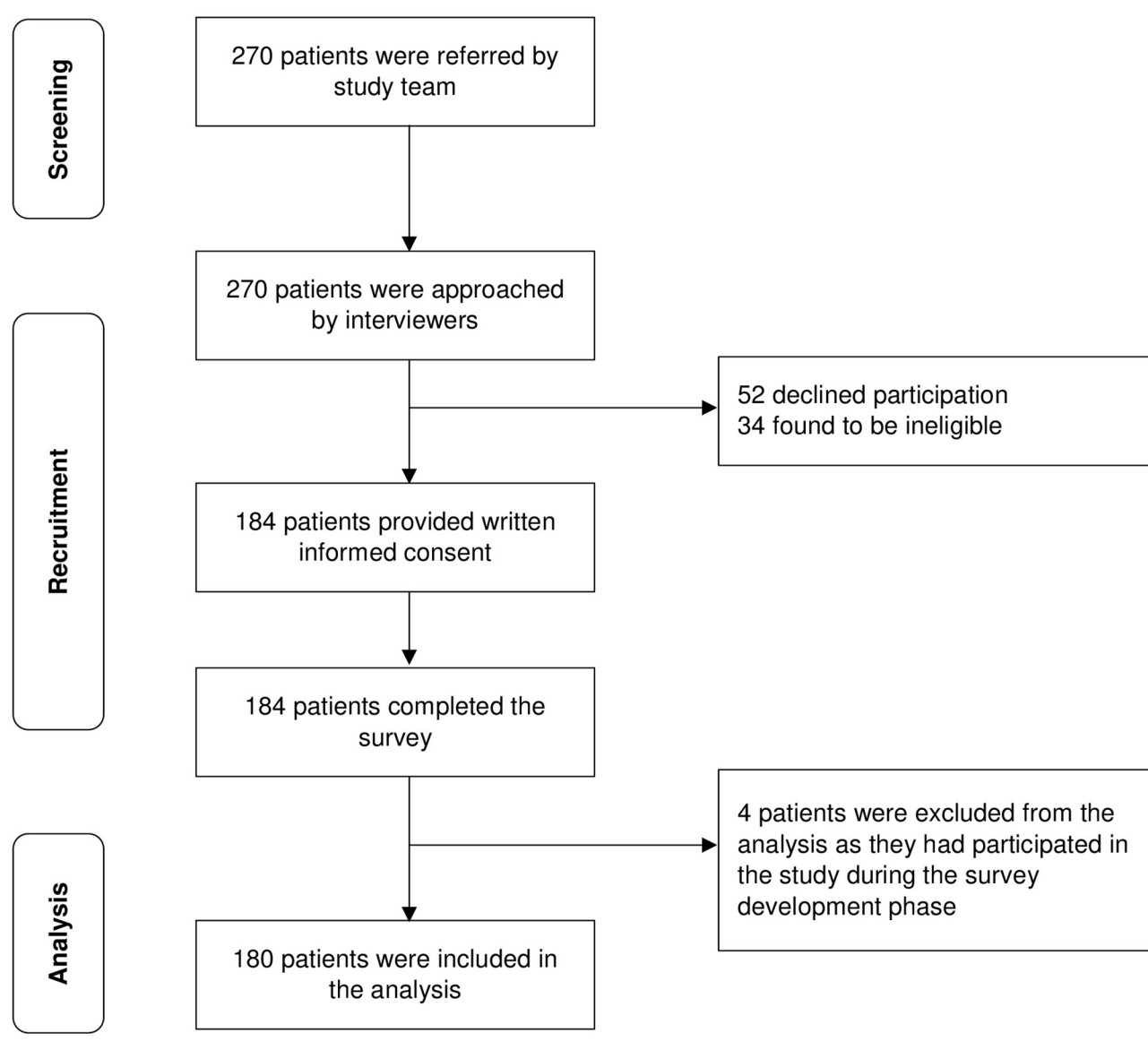

**Fig 1. Recruitment flowchart.**

Treatment B). Subsequently, participants were asked to choose between their preferred option (Treatment A or B) and their current treatment. Each treatment was defined by the above-mentioned attributes. The treatment options differed by the levels associated with each attribute. A sample choice task is shown in Fig 2.

## Experimental design

The experimental design was generated based on optimal D-efficiency using SAS 9.4. To assess if participants were paying attention, we created an attention choice task with one clear dominant alternative with better levels across all attributes than the other alternative. Choosing a non-dominant alternative may reflect patients' inattentiveness to the DCE task as we would expect a utility-maximizing individual to choose the dominant alternative [27]. To reduce

**Table 1. Attributes and levels included in the DCE.**

| Attributes | Levels |
|---|---|
| **1. Number of visits in a year** | |
| The number of visits to the eye clinic in a year may be different for each patient. These clinic visits may be for consultations only or for both consultations and injections. | 6 times a year<br>9 times a year<br>12 times a year |
| **2. Number of injections in a year** | |
| The number of injections a patient receives in a year may be different for each patient depending on the eye condition and the medicine. | 4 times a year<br>6 times a year<br>9 times a year<br>12 times a year |
| **3. Vision quality** | |
| Studies show that injections help improve the vision of patients with AMD. These vision improvements tend to occur in the first 3 to 4 months after starting injections and stay the same in most cases as long as the patient does not miss a clinic visit. | Good<br>Moderate<br>Poor |
| **4. Swelling in retina** | |
| Other than vision, your doctor would also be interested in knowing whether there is swelling in your retina. The scan of the retina shows whether there is swelling. Ideally there should be no swelling in the retina. Treatments can help control swelling. However, without treatment, the swelling of the retina can be expected to increase over time. | Well-controlled swelling<br>Moderately-controlled swelling<br>Poorly-controlled swelling |
| **5. Drug label** | |
| Drugs have to go through clinical trials to test whether they work and are safe before they can be approved for use for a specific condition.<br>• On-label drugs are those that are approved for use for AMD. They are also specifically packaged for treating AMD.<br>• Off-label drugs are those which are approved for other diseases but used to treat AMD. Healthcare provider needs to take out the medicine and put it in the injection manually. This process may increase the chances of infections. | On-label drugs<br>Off-label drugs |
| **6. Yearly out-of-pocket cost** | |
| Out-of-pocket cost refers to the total amount you or your family have to pay in a year for all treatment related costs, including costs for eye tests, injections and consultations after deductions from your insurance and other subsidies. | SGD 150/injection<br>SGD 400/injection<br>SGD 800/injection<br>SGD 1,500/injection<br>The costs shown to the respondents were annual total cost = cost per injection * number of injections in a year |

cognitive burden, the total number of choice tasks were divided into three blocks. Patients were randomized to only one of the blocks. Each patient was requested to answer a total of eight choice tasks (which includes the attention-testing task).

## Construction of final survey

The questionnaire first provided background information about nAMD and explained each attribute and the accompanying levels. The questionnaire also asked about patients' experience with the disease and existing medication. Instructions on how to answer the DCE tasks were then provided. The questionnaire included socio-demographic questions and the Brief Impact of Vision Impairment (B_IVI) scale. This validated 15-item instrument provides an overall

**B8. If you had to choose one of treatments, which would you choose?**

| | Treatment A | Treatment B |
|---|---|---|
| Number of total clinic visits in a year | 6 times a year | 9 times a year |
| Number of injections in a year | 4 times a year | 9 times a year |
| Vision quality | Good | Moderate |
| Swelling in retina | Well-controlled swelling | Moderately-controlled swelling |
| Drug label | On-label | Off-label |
| Out-of-pocket cost for all visits in a year | $6,000/year | $1,350/year |
| | Treatment A ☐ | Treatment B ☐ |

**B8.1. Comparing between treatment _____ and your current treatment, which would you choose?**

☐₁ Treatment ___

☐₂ My current treatment

**Fig 2. Sample choice task.**

measurement on patient's visual-related quality of life. It has two subscales that examine visual functioning and emotional well-being [28]. Scores for the visual functional, emotional well-being, and overall B_IVI_scale range from 1 to 4. Lower scores are indicative of increased restriction of participation in daily activities due to vision impairment.

## Statistical analysis

According to Orme's formula [29,30], this study required a minimum sample of 167 patients. However, we recruited 180 patients to increase precision of estimates. We used a latent class logistic (LCL) model which allows the identification of 2 or more groups of respondents with similar preferences within the group but different preferences between groups [31].

We investigated the predictors of class membership. Potential predictors, which were identified through literature, included age, years on medication, fear of injections, how well patients were able to cover the cost of their nAMD medication (i.e., affordability) and B_IVI scores. These variables were included in the model one at a time. Only significant variables (p<0.05) were retained to keep the model parsimonious.

The final outcome in the DCE was indicated as a single choice among the 3 options in each choice task (Treatment A, Treatment B or patient's current treatment). The independent variables were the attribute levels. All attribute levels were dummy coded except for cost which was assumed to be linear. The worst level for each attribute was set as the reference level with a value of 0. The coefficients can be interpreted as preference weights for each attribute level, informing how important changes in one attribute is (e.g., decreases in the number of injections) while holding other attributes used in the design (e.g., vision quality) constant.

The model also had an alternative specific constant (ASC) for the current treatment reflecting the utility gained associated with the current treatment over the hypothetical treatment alternatives. The attribute levels for the current treatment were identified based on the answers to the questions in the questionnaire. For example, for the number of clinic visits and number of injections for current treatment, we assigned the level that was closest to the number reported by the participants. If participants reported being unaware of drug labelling and control of swelling in their retina, we assigned "off-label" and "poorly-controlled swelling", respectively. If they were unaware of the out-of-pocket cost, they were assigned to the median cost reported by the other respondents for their current treatment. We conducted sensitivity analysis on the assignment of the attribute levels for the current treatment by varying each assumption individually. More information on this and DCE analysis are provided in S1 File.

Using the preference weights from the LCL model, we calculated the relative attribute importance (RAI) for each attribute for each class [32]. The difference between the best and worst coefficients of an attribute can be interpreted as the change in utility from the least to most desirable attribute level. The greater this difference is, the more important that attribute is, compared to the other attributes. We scaled RAI to present it as a proportion out of 100 for each class. We then calculated the RAI for the overall sample by weighing the RAI for each class by their representation in the sample. The RAI results can only be interpreted within the attributes and levels used in this study. The choice data was analyzed using Nlogit 6 while Stata 15.1 was used for descriptive statistics.

## Results

### Patient characteristics

Table 2 presents the patient characteristics. The mean age of participants was 71.6±0.7. The majority were male (61.1%), Chinese (90.6%), and married (69.4%). About 69.4% had primary or secondary education while 8.9% did not have a formal education. Only 31.1% had full-time or part-time jobs.

Over half of the patients (57.8%) reported receiving injections for more than 1 year. The mean number of eye clinic visits in the past year was 7.4±3.3. Patients reported receiving 5.9 ±3.0 injections. The vast majority of patients (91.7%) reported never having missed any

**Table 2.** Participant characteristics (N = 180).

| Characteristic | n = 180 |
|---|---|
| **Age, mean (SD), year** | 71.6 (0.7) |
| **Sex, No. (%)** | |
| Male | 110 (61.1%) |
| Female | 70 (38.9%) |
| **Ethnicity, No. (%)** | |
| Chinese | 163 (90.6%) |
| Malay | 8 (4.4%) |
| Indian | 9 (5.0%) |
| **Marital Status, No. (%)** | |
| Married | 125 (69.4%) |
| Single | 55 (30.6%) |
| **Education, No. (%)** | |
| No formal education | 16 (8.9%) |
| Primary | 65 (36.1%) |
| Secondary | 60 (33.3%) |
| Vocational/ITE | 5 (2.8%) |
| A levels/Polytechnic/Diploma | 16 (8.9%) |
| University and above | 18 (10.0%) |
| **Employment status, No. (%)** | |
| Full-time employment | 34 (18.9%) |
| Part-time employment | 22 (12.2%) |
| Not employed | 124 (68.9%) |
| **Anti-VEGF prescription history, No. (%)** | |
| Equal to or less than 1 year | 76 (42.2%) |
| Greater than 1 year | 104 (57.8%) |
| **Number of visits to eye clinic in the past year, mean (SD)** | 7.4 (3.3) |
| **Number of anti-VEGF injections received in the past year, mean (SD)** | 5.9 (3.0) |
| **Self-reported adherence to anti-VEGF treatment in the past year, No.(%)** | |
| Never missed a scheduled visit | 165 (91.7%) |
| Ever missed a scheduled visit | 15 (8.3%) |
| **Self-reported anxiety associated with intravitreal injections, No. (%)** | |
| Very anxious | 27 (15.0%) |
| A little anxious | 61 (33.9%) |
| Not anxious | 92 (51.1%) |
| **Self-perceived pain associated with intravitreal injection, No. (%)** | |
| Very painful | 11 (6.1%) |
| A little painful | 116 (64.4%) |
| Not painful | 53 (29.4%) |
| **Brief impact of vision impairment (B_IVI), mean (SD)** | |
| Overall score; Ranges: 1–4 | 3.5 (0.4) |
| Visual function; Ranges: 1–4 | 3.5 (0.5) |
| Emotional well-being; Ranges: 1–4 | 3.5 (0.4) |
| **Self-reported vision level, No. (%)** | |
| Good | 132 (73.3%) |
| Moderate | 45 (25.0%) |
| Poor | 3 (1.7%) |
| **Self-reported retina scan outcome, No. (%)** | |

*(Continued)*

**Table 2.** (Continued)

| Characteristic | n = 180 |
|---|---|
| Well-controlled | 55 (30.6%) |
| Moderately controlled | 94 (52.2%) |
| Poorly controlled | 10 (5.6%) |
| Not sure | 21 (11.7%) |
| **Self-reported type of medication label, No. (%)** | |
| On-label | 56 (31.1%) |
| Off-label | 82 (45.6%) |
| Not sure | 42 (23.3%) |
| **Self-reported ability to cover the out-of-pocket medical cost, No. (%)** | |
| Not paying for medication cost | 47 (26.1%) |
| Able to cover the cost very well | 7 (3.9%) |
| Able to cover the cost fairly well | 94 (52.2%) |
| Cost was poorly covered | 32 (17.8%) |

Percentages rounded to the nearest tenth and therefore each category may not sum up to 100%.

SD: Standard Deviation.

ITE: Institute of Technical Education.

scheduled visits. One-third (33.9%) reported getting a little anxious and 15.0% were very anxious about receiving intravitreal injections. The majority (64.4%) reported that the intravitreal injections were a little painful while 6.1% reported that it was very painful. The rest (29.4%) reported no pain. The overall B_IVI score was 3.5±0.4 while the visual function and emotional well-being subscale scores were 3.5±0.5 and 3.5±0.4, respectively.

Most patients (73.3%) reported having good vision while only 1.7% reported having poor vision. About half (52.2%) reported that their most recent scan showed moderately-controlled swelling while one-third (30.6%) reported well-controlled swelling and 11.7% were unsure. Most patients (45.6%) reported using off-label medications. About one-third (31.1%) reported using on-label medication while 23.3% were unsure. About a quarter (26.1%) reported not having to pay any out-of-pocket cost while 3.9% reported being able to cover the out-of-pocket cost very well.

## Treatment preferences

No patient failed the attention test. Among 2, 3, and 4-class LCL models, the 2-class LCL model was chosen based on the significance of estimates, the number of low prevalence classes, and Akaike information criterion. Classes 1 and 2 constituted 56.5% and 43.5% the sample (Table 3).

Both classes preferred better vision and lower medical costs. They also preferred moderately-controlled swelling over poorly-controlled swelling and on-label medication over off-label medication. Although the injection frequency did not affect preferences, both classes preferred 6 clinic visits to more frequent number of clinic visits. The main difference between the classes was their preference for their current treatment: Class 1, on average, had positive preferences for their current treatment over alternative treatments ($\beta = 1.65$, p-value<0.01), holding all else equal, whereas it was the opposite for Class 2 ($\beta = -1.83$, p-value<0.01).

Fig 3 shows the RAI for both classes and overall sample. For Class 1, out-of-pocket cost (33%) and vision quality (30%) were the most important attributes while vision quality was the most important attribute for Class 2 (40%), followed by number of total clinic visits (18%).

**Table 3. Latent class logistic regression results (2 classes).**

| | Class 1 | | | Class 2 | | |
|---|---|---|---|---|---|---|
| | Coefficient | Standard error | P-value | Coefficient | Standard error | P-value |
| **Number of visits in a year** | | | | | | |
| 6 times in a year | 1.6* | 0.3 | 0.00 | 2.2* | 0.4 | 0.00 |
| 9 times in a year | 0.3 | 0.2 | 0.23 | 0.3 | 0.3 | 0.27 |
| 12 times in a year[†] | 0 | | | 0 | | |
| **Number of injections in a year** | | | | | | |
| 4 times in a year | -0.1 | 0.4 | 0.85 | 0.3 | 0.5 | 0.58 |
| 6 times in a year | -0.3 | 0.4 | 0.45 | 0.3 | 0.6 | 0.60 |
| 9 times in a year | 0.4 | 0.3 | 0.22 | 0.6 | 0.5 | 0.26 |
| 12 times in a year[†] | 0 | | | 0 | | |
| **Vision quality** | | | | | | |
| Good | 3.8* | 0.4 | 0.00 | 4.9* | 0.5 | 0.00 |
| Moderate | 2.4* | 0.4 | 0.00 | 2.8* | 0.4 | 0.00 |
| Poor[†] | 0 | | | 0 | | |
| **Swelling in retina** | | | | | | |
| Well-controlled swelling | 0.6* | 0.3 | 0.04 | 0.4 | 0.3 | 0.20 |
| Moderately controlled swelling | 1.1* | 0.3 | 0.00 | 1.7* | 0.4 | 0.00 |
| Poorly controlled swelling[†] | 0 | | | 0 | | |
| **Drug label** | | | | | | |
| On-label drugs | 1.6* | 0.2 | 0.00 | 1.3* | 0.2 | 0.00 |
| Off-label drugs[†] | 0 | | | 0 | | |
| **Yearly out-of-pocket cost** | | | | | | |
| | -0.3* | 0.0 | 0.00 | -0.1* | 0.0 | 0.04 |
| **Alternative specific constant (ASC) for current treatment** | | | | | | |
| ASC for current treatment | 1.65* | 0.3 | 0.00 | -1.83* | 0.3 | 0.00 |
| **Class 1 membership predictors** | | | | | | |
| Being on injections for more than 1 year (Ref: Being on injections for less than 1 year) | 2.8* | 0.6 | 0.00 | | | |
| Able to cover the medication costs very well or do not pay for medication costs (Ref: Cover costs fairly well or poorly) | -1.3 | 0.8 | 0.13 | | | |
| Constant | -0.2 | 0.8 | 0.76 | | | |

* indicates p<0.05.

† indicates the reference attribute level.

These were followed by control of swelling in the retina (8% and 14% for Classes 1 and 2) and drug labelling (13% and 11% for Classes 1 and 2). The least important attribute was injection frequency for both classes (3% and 5%). For the overall sample, vision quality (34%) was the most important attribute, followed by out-of-pocket cost (24%). They were followed by number of total clinic visits (15%), labelling (12%) and control of swelling in the retina (11%). The injection frequency was the least important (4%).

We found only two significant predictors for class membership. Patients who reported being on injections for at least 1 year (compared to those on injections less than 1 year) were more likely to be in Class 1 (β = 2.84, p-value<0.01), and those who were able to cover their medication costs very well or do not pay their medication costs (compared to those who can cover their costs fairly well or poorly) were less likely to be in Class 1 (β = -1.27, p-value = 0.13). However, when we entered these two variables into the model, affordability was

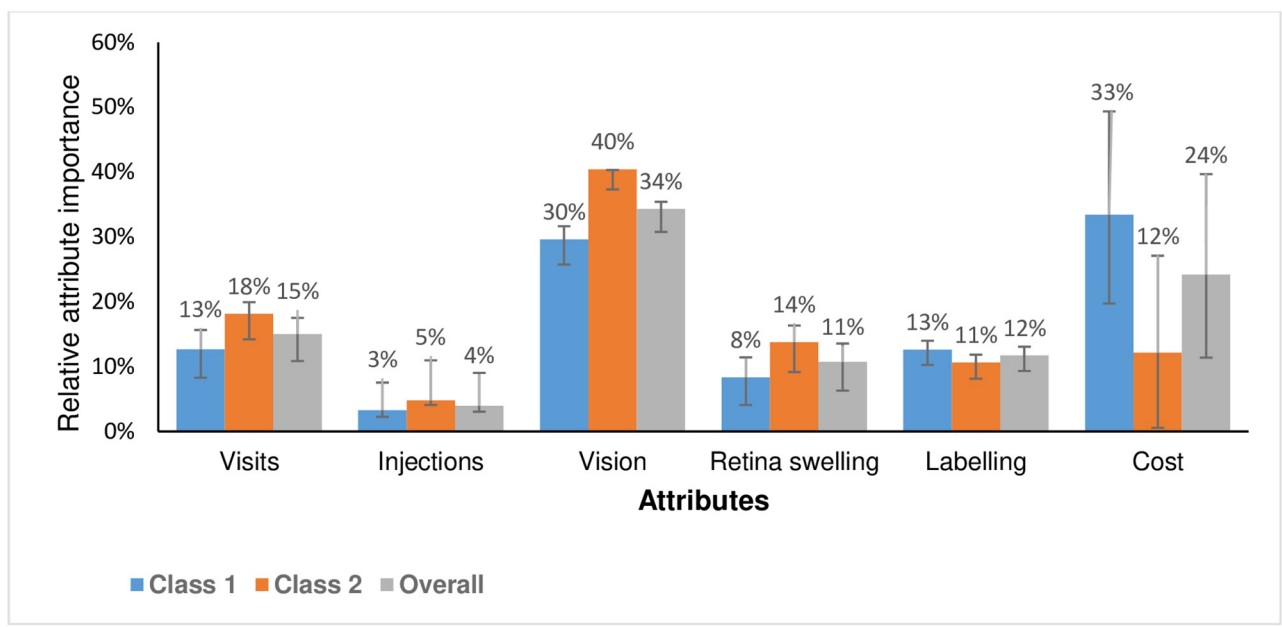

**Fig 3. Relative attribute importance (out of 100%) for Class 1, Class 2, and overall sample.**

not significant in predicting class membership. Age, anxiety about injections, fear of injections, and B_IVI scores were also found to be not significant.

As part of the sensitivity analyses, we conducted seven different variations in how we assigned the attributes levels for the current treatment. The differences in the RAI were mostly 1 or 2 percentage points, with the largest change of 3 percentage-points for the control of swelling in the retina for Class 1. These findings confirmed the robustness of the model.

## Discussion

Compared to current anti-VEGF agents (e.g. aflibercept), new nAMD therapies (e.g., brolu-ciuzmab, faricimab) have been reported to achieve non-inferior visual outcomes, and reported a higher proportion of eyes achieved fluid-free retina and offer reduced need for retreatment due to longer durability [12,33]. To understand patient preferences for anti-VEGF treatments, we quantified the importance of convenience factors against visual and anatomical outcomes, drug labelling, and out-of-pocket medical costs. Among the attributes evaluated, vision quality was the most important for the overall sample (34%), followed by the out-of-pocket medical costs (24%). Frequency of total clinic visits (15%), drug labelling (12%), and anatomical outcome of swelling in the retina (11%) were only of some importance to the patients. Given the same treatment outcomes and medical costs, patients preferred fewer total clinic visits while the injection frequency alone did not influence patient treatment preferences and was the least important attribute (4%).

The latent class model showed that there were two groups of patients with distinct preferences. Each group constituted about half of the patients (56.5% for Class 1 and 43.5% for Class 2). The main difference between both groups was their preference towards their current treatment. Class 1, on average, had positive preferences for their current treatment over alternative treatments whereas it was the opposite for Class 2. Patients with positive preferences for their current treatment were more likely to be on their current treatment for more than one year. This might be because these patients responded well to their treatment regimen and preferred

to stay on it. On the other hand, those who had been on a treatment regimen for less than one year (at the time of the survey) might not have yet observed the benefits of treatment.

The RAI were slightly different between both classes. For Class 1 patients, the most important attributes were out-of-pocket cost (33%) and vision quality (30%). Class 1 patients were also more likely to report that they could cover their current medical expenses only fairly-well or poorly (versus very well). This might explain why out-of-pocket cost was found to be the most important attribute for Class 1 patients. Our findings suggest that affordability could be an issue for these patients, and they would be sensitive to the slight changes in medication cost. For Class 2 patients, vision quality was the most important attribute (40%).

The total number of clinic visits (including visits for injections) was the third most important attribute for Class 1 (13%) and second most important attribute for Class 2 (18%). This finding suggests that convenience is important to the patients in our sample, albeit less compared to vision quality (for both classes) and cost (for Class 1 only). Given the age of these patients, clinic visits are likely to be burdensome and time consuming for both patients and their caregivers. We also found that our sample did not consider injection frequency to be important. There may be several reasons for this. First, since patients were most interested in controlling their vision quality, they may accept injections as necessary to improve their outcomes [34]. Second, our sample included patients who were already on intravitreal injections—they may therefore not be as fearful of injections compared to injection-naïve patients. Our findings also show that most patients found the injection-associated pain and anxiety to be bearable. Third, given that patients expected to receive injections, the frequency of injections may be unimportant to them. Fourth, the injection frequency could be confounded by the number of total clinic visits.

Although a safety attribute was not included in our study design, this was considered through the drug-labelling attribute. Drug-labelling indicates that on-label medications are approved for nAMD treatment based on their safety profile while safety has not necessarily been established for off-label use of medications [35,36]. Drug-labelling had some impact on preferences when the medication cost (and other attributes) was held constant. As on-label medications tend to be more expensive than off-label medications, and out-of-pocket cost was much more important to the patients in our sample, the preference for an on-label medication is likely to be influenced by how much more expensive it is compared to off-label medications.

Our study had several strengths. First, the use of a DCE allowed us to systematically investigate how important changes in one attribute is while holding other attributes constant. Second, it also allowed us to quantify the importance of one attribute relative to the other attributes based on the attributes and levels used in the study. Third, this is one of the only DCE studies to investigate the importance of out-of-pocket medical costs against other outcomes, and the finding showed that out-of-pocket cost could be a major concern (i.e., the most important attribute) for about half of the patients in our sample. Our study also had several limitations. As only patients who were already on injections were sampled, our results cannot be generalized to newly diagnosed injection-naïve patients as fear of injections or injection frequency may have a larger impact on the preferences of these patients. Also, most of the participants in this study reported high treatment adherence, thus limiting the generalizability of results to those with poor adherence. We also used a convenience sampling of patients from a single institution. Although patients in Singapore make out-of-pocket payments, they also receive a complex mix of subsidies. Results therefore may not be generalizable to other healthcare systems, especially those where costs are not charged at the point of service.

Our study has several important implications. The most important attribute was vision quality. Given the same vision quality outcomes, patients preferred treatment regimens which required fewer total clinic visits. This suggests that it may be important to consider new models

of decentralized care, where patients with poor mobility and multiple co-morbidities can be offered treatment at a more accessible location. The retreatment frequency also affected out-of-pocket medical costs, a major concern for some patients in our sample. Given the multiple anti-VEGF treatment options with similar benefits, patient preferences should be assessed and incorporated while choosing a treatment regimen to improve patient's treatment acceptance. This may lead to a more effective AMD management which can improve long-term treatment adherence, translating to better patient outcomes.

## Supporting information

**S1 File. Statistical analysis.**
(DOCX)

## Author Contributions

**Conceptualization:** Semra Ozdemir, Eric Finkelstein, Jia Jia Lee, Issac Horng Khit Too, Tien Yin Wong, Gemmy Chui Ming Cheung.

**Data curation:** Jia Jia Lee, Kelvin Yi Chong Teo, Anna Chen Sim Tan, Tien Yin Wong.

**Formal analysis:** Semra Ozdemir.

**Funding acquisition:** Semra Ozdemir.

**Methodology:** Semra Ozdemir, Eric Finkelstein.

**Project administration:** Jia Jia Lee, Kelvin Yi Chong Teo, Anna Chen Sim Tan, Tien Yin Wong.

**Writing – original draft:** Semra Ozdemir, Jia Jia Lee.

**Writing – review & editing:** Semra Ozdemir, Eric Finkelstein, Jia Jia Lee, Issac Horng Khit Too, Kelvin Yi Chong Teo, Anna Chen Sim Tan, Tien Yin Wong, Gemmy Chui Ming Cheung.

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
