## [Decision Letter · Decision Letter 0]

3 May 2022

PONE-D-21-35970Understanding patient preferences in anti-VEGF treatment options for age-related macular degenerationPLOS ONE

Dear Dr. Ozdemir,

Thank you for submitting your manuscript to PLOS ONE. After careful consideration, we feel that it has merit but does not fully meet PLOS ONE’s publication criteria as it currently stands. Therefore, we invite you to submit a revised version of the manuscript that addresses the points raised during the review process.

We look forward to receiving your revised manuscript.

Kind regards,

Marie-Helene Errera

Academic Editor

PLOS ONE

“This study was funded by Novartis (Singapore). The funder participated in the design of the study, the approval of the manuscript, and the decision to submit the manuscript for publication. The funder had no role in the collection, management, analysis, and interpretation of the data; preparation, and review of the manuscript.”

“This study was funded by Novartis (Singapore). The funder participated in the design of the study, the approval of the manuscript, and the decision to submit the manuscript for publication. The funder had no role in the collection, management, analysis, and interpretation of the data; preparation, and review of the manuscript.”

“Semra Ozdemir, Anna Cheng Sim Tan, Kelvin Yi Chong Teo, Tien Yin Wong and Gemmy Chui Ming Cheung reported receiving grants from Novartis during the conduct of the study. Issac Horng Khit Too is being employed by Novartis. Eric Finkelstein and Jia Jia Lee reported no conflict of interest.”

6. We note that you have indicated that data from this study are available upon request. PLOS only allows data to be available upon request if there are legal or ethical restrictions on sharing data publicly. For more information on unacceptable data access restrictions, please see http://journals.plos.org/plosone/s/data-availability#loc-unacceptable-data-access-restrictions.

7. PLOS requires an ORCID iD for the corresponding author in Editorial Manager on papers submitted after December 6th, 2016. Please ensure that you have an ORCID iD and that it is validated in Editorial Manager. To do this, go to ‘Update my Information’ (in the upper left-hand corner of the main menu), and click on the Fetch/Validate link next to the ORCID field. This will take you to the ORCID site and allow you to create a new iD or authenticate a pre-existing iD in Editorial Manager. Please see the following video for instructions on linking an ORCID iD to your Editorial Manager account: https://www.youtube.com/watch?v=_xcclfuvtxQ.

...

Dear Dr. Ozdemir,

This is an interesting paper. Please Review and address the comments from Reviewers #1 and #2.

Some passages might benefit from professional language editing. Please contact a professional language editor in order to address this issue.

My own comments are below:

Page 20 line 240. The reader does not really understand what means Class 2,1 etc… when not familiar with the LCL model. Would it be possible to explain a little bit to what these classes correspond to “subset of Pts with comparable answers to question…” “subgroup of patients with ….” Clusters of patients based on the similar questions answers ….. or who have similar …”

I see that you explain it later in Discussion line 292 “We found two groups of patients with distinct preferences.” You should explain it earlier as well to facilitate the reader work.

“Among 2, 3, and 4-class LCL models, the 2-class LCL model was chosen based on the significance of estimates, the number of low prevalence classes, and Akaike information criterion. Classes 1 and 2 constituted 56.5% and 43.5% the sample.”

First line of Discussion, “Compared to current anti-VEGF agents (e.g. aflibercept), New nAMD therapies (e.g., broluciuzmab,…” change for “new”.

Page 34 line 321 What do you mean by “We found two groups of patients with distinct preferences.”

Line 327: This sentence is unclear . “As out-of-pocket was much more important to the patients, the preference for an on-label medication is likely to be influenced by its relative price.”

Reviewers' comments:

**Comments to the Author**

1. Is the manuscript technically sound, and do the data support the conclusions?

Reviewer #1: Yes

Reviewer #2: Yes

2. Has the statistical analysis been performed appropriately and rigorously? 

Reviewer #1: Yes

Reviewer #2: I Don't Know

3. Have the authors made all data underlying the findings in their manuscript fully available?

Reviewer #1: Yes

Reviewer #2: No

4. Is the manuscript presented in an intelligible fashion and written in standard English?

Reviewer #1: Yes

Reviewer #2: Yes

5. Review Comments to the Author

Reviewer #1: The authors present an interesting paper to help understand patient preferences for anti-VEGF therapy when they have neovascular AMD (nAMD). The authors found that patients prefer to have good vision as first preference followed by out of pocket cost as the most important factors in therapy. Other things like frequency of visits was third but not nearly as important as the other factors. They authors discovered this by means of a discrete choice experiment test they administered to their patients. Overall it is a well written and interesting paper. There are a few issues below to be rectified.

The main issue is that the authors split the data into classes 1 and classes 2. The authors should introduce the differences of classes 1 and 2 starting on line 244. The differences are explained starting on line 266. This should also be clarified to make the reasoning for the split more apparent.

In Table 2, I would reword Anti-VEGF prescription history to: Less than 1 year of treatment and Greater than 1 year of treatment.

More specific comments are below.

In the Introduction, line 59 aging is misspelled.

In the introduction, the authors mention that PRN treatment have demonstrated good outcomes as a treatment regimen for nAMD. Typically patients treated with a PRN fashion start to lose vision at the 2 year mark (HARBOR data) when compared with fixed interval dosing and patients in the SEVEN-UP study, at 7 years treated mainly by PRN did much worse than other studies that looked at fixed interval or treat and extend regimens. While PRN treatment is an option, it is inferior to fixed interval and treat and extend. The authors should clarify this point.

In the introduction, lines 68-70: This sentence should be reworded. Consider " These treatment regimens impose logistical and/or financial burden to the patients and their caregivers that often accompany the patients."

In the introduction, line 98-100, consider changing the sentence to "The results of this study may inform physicians on patient..."

On line 137, consider adding equivalent USD, Euros or Pounds to help international audiences to have an idea of cost.

On Table 2 , in the explanation, the authors mention that the percentages have been rounded to the nearest whole number. This is not correct. The authors have rounded the percentages to the nearest tenth. Correct this.

On line 223, change the sentence to " The vast majority of patients...

On line 226, remove the word "vast"

On line 241, consider changing the sentence to "No patients failed the attention test."

On line 298-300, This sentence should be reworded and clarified.

On line 320, the authors mention that injection frequency could be confounded by clinic visits. Another alternative that should be mentioned is that once patients are seen and evaluated, getting an intravitreal injection was already expected, so having another injection or not, was not a major factor.

On line 335, the authors mention that for about half of the patients in the sample, cost was most important attribute, but according to their percentages, it would be about one quarter of patients, and not total cost, but out of pocket cost. This should be clarified.

On line 347-349, the authors talk about home-monitoring, nothing in their study examined that variable and this conclusion cannot be drawn from their study. I would remove this sentence.

On line 349, "decentralized" is misspelled

Reviewer #2: Thank you very much for giving me the opportunity to review this interesting patient related outcome study. Ozdemir conducted a cross-sectional survey to identify which factors are most important for patients using established statistical techniques. Vision quality seemed to be the most important attribute, followed by out-of-pocket cost and convenience. Potential conflicts of interest have been clearly declared.

Overall, the study has been thoroughly designed and diligently conducted. There are some typos and some passages might benefit from professional language editing.

Minor issues:

Methods in abstract: bullet (3) should read >=21 years old, not <=

TnE is not a commonly used abbreviation for ‘treat-and-extend’.

6. PLOS authors have the option to publish the peer review history of their article (what does this mean?). If published, this will include your full peer review and any attached files.

Reviewer #1: **Yes: **Sean Adrean

Reviewer #2: No

---

## [Author Response · Author response to Decision Letter 0]

26 Jun 2022

17-May-2022

To,

Dr. Marie-Helene Errera

Academic Editor 

Dear Dr. Marie-Helene Errera,

Thank you for giving us the opportunity to revise and resubmit our manuscript. Please see our responses below. The blue font indicates the revised or newly added text.

Comments from the Editor:

Comment 1: This is an interesting paper. Please Review and address the comments from Reviewers #1 and #2. 

Response 1: Thank you. We have reviewed and addressed the comments from both reviewers. Please see below for our specific comments.

Comment 2: Some passages might benefit from professional language editing. Please contact a professional language editor in order to address this issue.

Response 2: We had a language editor review and revise the manuscript.

Comment 3: Page 20 line 240. The reader does not really understand what means Class 2,1 etc… when not familiar with the LCL model. Would it be possible to explain a little bit to what these classes correspond to “subset of Pts with comparable answers to question…” “subgroup of patients with ….” Clusters of patients based on the similar questions answers ….. or who have similar …”

I see that you explain it later in Discussion line 292 “We found two groups of patients with distinct preferences.” You should explain it earlier as well to facilitate the reader work.

Response 3: We added an explanation of the latent class model when we first mention it in the manuscript.

Statistical Analysis

“We used a latent class logistic (LCL) model which allows identifying 2 or more groups of respondents with similar preferences within the group but different preferences between groups.(31)”

Comment 4: First line of Discussion, “Compared to current anti-VEGF agents (e.g. aflibercept), New nAMD therapies (e.g., broluciuzmab,…” change for “new”.

Response 4: We have now revised it as “new”.

Comment 5: Page 34 line 321 What do you mean by “We found two groups of patients with distinct preferences.”

Response 5: This is the outcome of the latent class model. We have now revised the sentence to clarify this.

Discussion

“The latent class model showed that there were two groups of patients with distinct preferences.”

Comment 6: Line 327: This sentence is unclear. “As out-of-pocket was much more important to the patients, the preference for an on-label medication is likely to be influenced by its relative price.”

Response 6: Patients preferred on-label medications over off-label medications. However, since on-label medications tend to be much more expensive and out-of-pocket cost is much more important to the patients, the cost will drive the medication choice. We have revised the sentence to clarify this.

Discussion

“Drug-labelling had some impact on preferences when the medication cost (and other attributes) was held constant. As on-label medications tend to be more expensive than off-label medications, and out-of-pocket cost was much more important to the patients in our sample, the preference for an on-label medication is likely to be influenced by how much more expensive it is compared to off-label medications.”

Reviewers' Comments to the Author

Reviewer #1: 

The authors present an interesting paper to help understand patient preferences for anti-VEGF therapy when they have neovascular AMD (nAMD). The authors found that patients prefer to have good vision as first preference followed by out of pocket cost as the most important factors in therapy. Other things like frequency of visits was third but not nearly as important as the other factors. They authors discovered this by means of a discrete choice experiment test they administered to their patients. Overall it is a well written and interesting paper. There are a few issues below to be rectified.

Comment 7: The main issue is that the authors split the data into classes 1 and classes 2. The authors should introduce the differences of classes 1 and 2 starting on line 244. The differences are explained starting on line 266. This should also be clarified to make the reasoning for the split more apparent.

Response 7: We did not split the data into class 1 and class 2. The latent class model finds groups with similar preferences within the group based on respondents’ answers and splits these groups into unique classes. We have provided explanations to clarify this.

Statistical Analysis

“We used a latent class logistic (LCL) model which allows the identification of 2 or more groups of respondents with similar preferences within the group but different preferences between groups.(31)”

Discussion

“The latent class model showed that there were two groups of patients with distinct preferences.”

Comment 8: In Table 2, I would reword Anti-VEGF prescription history to: Less than 1 year of treatment and Greater than 1 year of treatment.

Response 8: We have revised the labels as “Equal to or less than 1 year” and “Greater than 1 year”.

Comment 9: In the Introduction, line 59 aging is misspelled.

Response 9: We have revised “ageing” with “aging”.

Comment 10: In the introduction, the authors mention that PRN treatment have demonstrated good outcomes as a treatment regimen for nAMD. Typically patients treated with a PRN fashion start to lose vision at the 2 year mark (HARBOR data) when compared with fixed interval dosing and patients in the SEVEN-UP study, at 7 years treated mainly by PRN did much worse than other studies that looked at fixed interval or treat and extend regimens. While PRN treatment is an option, it is inferior to fixed interval and treat and extend. The authors should clarify this point.

Response 10: We revised the introduction to clarify that the good outcomes of PRN are not maintained in the long term.

Introduction

“For the management of nAMD, the classic monthly or bi-monthly (fixed) dosing of the intravitreal injections of anti–vascular endothelial growth factor (anti-VEGF) has been proven efficacious and safe in clinical trials (4, 5) and has been used as standard care for nAMD in most countries.(6, 7) However, monthly or bi-monthly injections can be unsustainable in many real-world clinical practices. A more flexible approach, pro-re-nata (PRN) dosing (treat based on certain visual and anatomical criteria) has demonstrated comparable outcomes at 12 months but the visual gain is not sustained in the long term.(8, 9) The treat-and-extend approach (treating at every clinic visit where visits are extended until an ideal interval is established for each patient) has demonstrated good vision outcomes with fewer injections and/or clinical visits(10, 11) and have the potential to be more convenient and less expensive to patients and healthcare systems.” 

Comment 11: In the introduction, lines 68-70: This sentence should be reworded. Consider " These treatment regimens impose logistical and/or financial burden to the patients and their caregivers that often accompany the patients."

Response 11: We have revised the sentence.

Introduction

“Multiple anti-VEGF agents are increasingly becoming available for nAMD treatment, with newer agents (e.g., broluciuzmab, faricimab) potentially offering longer durability alongside fewer injections.(12-15) These treatment regimens impose different logistical and/or financial burden to patients and their informal caregivers who often accompany patients to the clinic visits.(16-18)”

Comment 12: In the introduction, line 98-100, consider changing the sentence to "The results of this study may inform physicians on patient..."

Response 12: We have revised the sentence.

Introduction

 “The results of this study may inform clinicians on patient preferences to facilitate shared decision-making, and healthcare policymakers for approval of new drugs.”

Comment 13: On line 137, consider adding equivalent USD, Euros or Pounds to help international audiences to have an idea of cost.

Response 13: We have added the USD equivalent values.

Establishing Attributes and Levels

The levels for the cost attribute were selected based on the medication costs (both unsubsidized and subsidized by the government based on means-testing) at local hospitals (which can range between SGD150-1,500 (~USD108-1080) per injection).

Comment 14: On Table 2 , in the explanation, the authors mention that the percentages have been rounded to the nearest whole number. This is not correct. The authors have rounded the percentages to the nearest tenth. Correct this.

Response 14: We have revised the sentence as the reviewer suggested.

Table 2

“Percentages rounded to the nearest tenth and therefore each category may not sum up to 100%.”

Comment 15: On line 223, change the sentence to " The vast majority of patients...

Response 15: We have revised the sentence.

Comment 16: On line 226, remove the word "vast"

Response 16: We have removed the word “vast”.

Comment 17: On line 241, consider changing the sentence to "No patients failed the attention test."

Response 17: We have revised the sentence as the reviewer suggested.

Comment 18: On line 298-300, This sentence should be reworded and clarified.

Response 18: We have revised the sentence.

Discussion

“Patients with positive preferences for their current treatment were more likely to be on their current treatment for more than one year. This might be because these patients responded well to their treatment regimen and preferred to stay on it. On the other hand, those who had been on a treatment regimen for less than one year (at the time of the survey) might not have yet observed the benefits of treatment.”

Comment 19: On line 320, the authors mention that injection frequency could be confounded by clinic visits. Another alternative that should be mentioned is that once patients are seen and evaluated, getting an intravitreal injection was already expected, so having another injection or not, was not a major factor.

Response 19: We have added the reviewer’s suggestion as an explanation.

Discussion

“We also found that our sample did not consider injection frequency to be important. There may be several reasons for this. First, since patients were most interested in controlling their vision quality, they may accept injections as necessary to improve their outcomes.(34) Second, our sample included patients who were already on intravitreal injections – they may therefore not be as fearful of injections compared to injection-naïve patients. Our findings also show that most patients found the injection-associated pain and anxiety to be bearable. Third, given that patients expected to receive injections, the frequency of injections may be unimportant to them. Fourth, the injection frequency could be confounded by the number of total clinic visits.”

Comment 20: On line 335, the authors mention that for about half of the patients in the sample, cost was most important attribute, but according to their percentages, it would be about one quarter of patients, and not total cost, but out of pocket cost. This should be clarified.

Response 20: We have revised “cost” as “out-of-pocket cost”. The out-of-pocket cost was the most important attribute for Class 1, which constitutes 56.5% of the sample. 

Discussion

“Third, this is one of the only DCE studies to investigate the importance of out-of-pocket medical costs against other outcomes, and the finding showed that out-of-pocket cost could be a major concern (i.e., the most important attribute) for about half of the patients in our sample.”

Comment 21: On line 347-349, the authors talk about home-monitoring, nothing in their study examined that variable and this conclusion cannot be drawn from their study. I would remove this sentence.

Response 21: We have removed any reference to “home monitoring” and revised this section. 

Discussion

“Given the same vision quality outcomes, patients preferred treatment regimens which require fewer total clinic visits. This suggests that it may be important to consider new models of decentralized care, where patients with poor mobility and multiple co-morbidities can be offered treatment at a more accessible location.”

Comment 22: On line 349, "decentralized" is misspelled.

Response 22: We have now changed the language to American English. The word "decentralised" is revised as “"decentralized".

Reviewer #2:

Comment 23: Thank you very much for giving me the opportunity to review this interesting patient related outcome study. Ozdemir conducted a cross-sectional survey to identify which factors are most important for patients using established statistical techniques. Vision quality seemed to be the most important attribute, followed by out-of-pocket cost and convenience. Potential conflicts of interest have been clearly declared.

Overall, the study has been thoroughly designed and diligently conducted. There are some typos and some passages might benefit from professional language editing.

Response 23: We had a language editor review and revise the manuscript.

Comment 24: Minor issues: Methods in abstract: bullet (3) should read >=21 years old, not <=

Response 24: We have revised the symbol to “>=”.

Comment 25: TnE is not a commonly used abbreviation for ‘treat-and-extend’.

Response 25: We have removed the acronym TnE.

---

## [Editor Report · Decision Letter 1]

18 Jul 2022

Understanding patient preferences in anti-VEGF treatment options for age-related macular degeneration

PONE-D-21-35970R1

Dear Dr. Ozdemir,

We’re pleased to inform you that your manuscript has been judged scientifically suitable for publication and will be formally accepted for publication once it meets all outstanding technical requirements.

Kind regards,

Marie-Helene Errera

Academic Editor

PLOS ONE
---

## [Editor Report · Acceptance letter]

3 Aug 2022

PONE-D-21-35970R1 

Understanding patient preferences in anti-VEGF treatment options for age-related macular degeneration 

Dear Dr. Ozdemir:

I'm pleased to inform you that your manuscript has been deemed suitable for publication in PLOS ONE. Congratulations! Your manuscript is now with our production department. 

Kind regards, 

on behalf of

Dr. Marie-Helene Errera 

Academic Editor

PLOS ONE